# Mobile phones: The effect of its presence on learning and memory

**Clarissa Theodora Tanil, Min Hooi Yong** *

Department of Psychology, Sunway University, Selangor, Malaysia

* mhyong@sunway.edu.my

## Abstract

Our aim was to examine the effect of a smartphone's presence on learning and memory among undergraduates. A total of 119 undergraduates completed a memory task and the Smartphone Addiction Scale (SAS). As predicted, those without smartphones had higher recall accuracy compared to those with smartphones. Results showed a significant negative relationship between phone conscious thought, "how often did you think about your phone", and memory recall but not for SAS and memory recall. Phone conscious thought significantly predicted memory accuracy. We found that the presence of a smartphone and high phone conscious thought affects one's memory learning and recall, indicating the negative effect of a smartphone proximity to our learning and memory.

**Data Availability Statement:** All relevant data are within the manuscript.

**Funding:** MHY received funding from Sunway University (GRTIN-RRO-104-2020 and INT-RRO-2018-49).

## Introduction

Smartphones are a popular communication form worldwide in this century and likely to remain as such, especially among adolescents [1]. The phone has evolved from basic communicative functions–calls only–to being a computer-replacement device, used for web browsing, games, instant communication on social media platforms, and work-related productivity tools, e.g. word processing. Smartphones undoubtedly keep us connected; however, many individuals are now obsessed with them [2,3]. This obsession can lead to detrimental cognitive functions and mood/affective states, but these effects are still highly debated among researchers.

Altmann, Trafton, and Hambrick suggested that as little as a 3-second distraction (e.g. reaching for a cell phone) is adequate to disrupt attention while performing a cognitive task [4]. This distraction is disadvantageous to subsequent cognitive tasks, creating more errors as the distraction period increases, and this is particularly evident in classroom settings. While teachers and parents are for [5] or against cell phones in classrooms [6], empirical evidence showed that students who used their phones in class took fewer notes [7] and had poorer overall academic performance, compared to those who did not [8,9]. Students often multitask in classrooms and even more so with smartphones in hand. One study showed no significant difference in in-class test scores, regardless of whether they were using instant messaging [10]. However, texters took a significantly longer time to complete the in-class test, suggesting that texters required more cognitive effort in memory recall [10]. Other researchers have posited

**Competing interests:** The authors have declared that no competing interests exist.

that simply the presence of a cell phone may have detrimental effects on learning and memory as well. Research has shown that a mobile phone left next to the participant while completing a task, is a powerful distractor even when not in use [11,12]. Their findings showed that mobile phone participants could perform similarly to control groups on simple versions of specific tasks (e.g. visual spatial search, digit cancellation), but performed much poorer in the demanding versions. In another study, researchers controlled for the location of the smartphone by taking the smartphones away from participants (low salience, LS), left the smartphone next to them (high salience/HS), or kept the smartphones in bags or pockets (control) [13]. Results showed that participants in LS condition performed significantly better compared to HS, while no difference was established between control and HS conditions. Taken together, these findings confirmed that the smartphone is a distractor even when not in use. Further, smartphone presence also increases cognitive load, because greater cognitive effort is required to inhibit distractions.

Reliance on smartphones has been linked to a form of psychological dependency, and this reliance has detrimental effect on our affective 'mood' states. For example, feelings of anxiety when one is separated from their smartphones can interfere with the ability to attend to information. Cheever et al. observed that heavy and moderate mobile phone users reported increased anxiety when their mobile phone was taken away as early as 10 minutes into the experiment [14]. They noted that high mobile phone usage was associated with higher risk of experiencing 'nomophobia' (no mobile phone phobia), a form of anxiety characterized by constantly thinking about one's own mobile phones and the desire to stay in contact with the device [15]. Other studies reported similar separation-anxiety and other unpleasant thoughts in participants when their smartphones were taken away [16] or the usage was prohibited [17,18]. Participants also reported having frequent thoughts about their smartphones, despite their device being out of sight briefly (kept in bags or pockets), to the point of disrupting their task performance [13]. Taken together, these findings suggest that strong attachment towards a smartphone has immediate and lasting negative effects on mood and appears to induce anxiety.

Further, we need to consider the relationship between cognition and emotion to understand how frequent mobile phone use affects memory e.g. memory consolidation. Some empirical findings have shown that anxious individuals have attentional biases toward threats and that these biases affect memory consolidation [19,20]. Further, emotion-cognition interaction affects efficiency of specific cognitive functions, and that one's affective state may enhance or hinder these functions rapidly, flexibly, and reversibly [21]. Studies have shown that positive affect improves visuospatial attention [22], sustained attention [23], and working memory [24]. The researchers attributed positive affect in participants' improved controlled cognitive processing and less inhibitory control. On the other hand, participants' negative affect had fewer spatial working memory errors [23] and higher cognitive failures [25]. Yet, in all of these studies–the direction of modulation, intensity, valence of experiencing a specific affective state ranged widely and primarily driven by external stimuli (i.e. participants affective states were induced from watching videos), which may not have the same motivational effect generated internally.

## Present study

Prior studies have demonstrated the detrimental effects of one's smartphone on cognitive function (e.g. working memory [13], visual spatial search [12], attention [11]), and decreased cognitive ability with increasing attachment to one's phone [14,16,26]. Further, past studies have demonstrated the effect of affective state on cognitive performance [19,20,22–25,27]. To

our knowledge, no study has investigated the effect of positive or negative affective states resulting from smartphone separation on memory recall accuracy. One study showed that participants reporting an increased level of anxiety as early as 10 minutes [14]. We also do not know the extent of smartphone addiction and phone conscious thought effects on memory recall accuracy. One in every four young adults is reported to have problematic smartphone use and this is accompanied by poor mental health e.g. higher anxiety, stress, depression [28]. One report showed that young adults reached for their phones 86 times in a day on average compared to 47 times in other age groups [29]. Young adults also reported that they "definitely" or "probably" used their phone too much, suggesting that they recognised their problematic smartphone use.

We had two main aims in this study. First, we replicated [13] to determine whether 'phone absent' (LS) participants had higher memory accuracy compared to the 'phone present' (HS). Second, we predicted that participants with higher smartphone addiction scores (SAS) and higher phone conscious thought were more likely to have lower memory accuracy. With regards to separation from their smartphone, we hypothesised that LS participants will experience an increase of negative affect or a decrease in positive affect and that this will affect memory recall negatively. We will also examine whether these predictor variables–smartphone addiction, phone conscious thought and affect differences—predict memory accuracy.

## Materials and methods

### Participants

A total of 119 undergraduate students (61 females, $M_{age}$ = 20.67 years, $SD_{age}$ = 2.44) were recruited from a private university in an Asian capital city. To qualify for this study, the participant must own a smartphone and does not have any visual or auditory deficiencies. Using G*Power v. 3.1.9.2 [30], we require at least 76 participants with an effect size of $d$ = .65, α = .05 and power of (1-β) = .8 based on Thornton et al.'s [11] study, or 128 participants from Ward's study [13].

Out of 119 participants, 43.7% reported using their smartphone mostly for social networking, followed by communication (31.1%) and entertainment (17.6%) (see Table 1 for full details on smartphone usage). Participants reported an average smartphone use of 8.16 hours in a day ($SD$ = 4.05). There was no significant difference between daily smartphone use for participants in the high salience (HS) and low salience groups (LS), $t$ (117) = 1.42, $p$ = .16, Cohen's $d$ = .26. Female participants spent more time using their smartphones over a 24-hour period ($M$ = 9.02, $SD$ = 4.10) compared to males, ($M$ = 7.26, $SD$ = 3.82), $t$ (117) = 2.42, $p$ = .02, Cohen's $d$ = .44.

### Ethical approval and informed consent

The study was conducted in accordance with the protocol approved by the Department of Psychology Research Ethics Committee at Sunway University (approval code: 20171090). All

**Table 1. Most frequently used phone feature ($n$ = 119).**

|  | $n$ | % |
|---|---|---|
| Social networking (Instagram, Twitter, Facebook) | 52 | 43.7 |
| Communication (WhatsApp, Line, messaging, calls, emails) | 37 | 31.1 |
| Entertainment (music, games, videos) | 21 | 17.6 |
| Web surfing | 8 | 6.7 |
| Productivity (camera, calculator, alarm, calendar) | 1 | 0.8 |

participants provided written consent before commencing the study and were not compensated for their participation in the study.

## Study design

Our experimental study was a mixed design, with smartphone presence (present vs absent) as a between-subjects factor, and memory task as a within-subjects factor. Participants who had their smartphone out of sight formed the 'Absent' or low-phone salience (LS) condition, and the other group had their smartphone placed next to them throughout the study, 'Present' or high-phone salience (HS) condition. The dependent variable was recall accuracy from the memory test.

## Stimuli

**Working memory span test.** A computerized memory span task 'Operation Span (OS)' retrieved from software Wadsworth CogLab 2.0 was used to assess working memory [31]. A working memory span test was chosen as a measure to test participants' memory ability for two reasons. First, participants were required to learn and memorize three types of stimuli thus making this task complex. Second, the duration of task completion took approximately 20 minutes. This was advantageous because we wanted to increase separation-anxiety [16] as well as having the most pronounced effect on learning and memory without the presence of their smartphone [9].

The test comprised of three stimulus types, namely words (long words such as computer, refrigerator and short words like pen, cup), letters (similar sound E, P, B, and non-similar sound D, H, L) and digits (1 to 9). The test began by showing a sequence of items on the left side of the screen, with each item presented for one second. After that, participants were required to recall the stimulus from a 9-button box located on the right side of the screen. In order to respond correctly, participants were required to click on the buttons for the items in the corresponding order they were presented. A correct response increases the length of stimulus presented by one item (for each stimulus category), while an incorrect response decreases the length of the stimulus by one item. Each trial began with five stimuli and increased or decreased depending on the participants' performance. The minimum length possible was one while the maximum was ten. Each test comprised of 25 trials with no time limit and without breaks between trials. Working memory ability was measured through the number of correct responses over total trials: scores ranged from 0 to 25, with the highest score representing superior working memory.

**Positive and Negative Affect Scale (PANAS).** We used PANAS to assess the current mood/affective state of the participants with state/feeling-descriptive statements [32]. PANAS has ten PA statements e.g. interested, enthusiastic, proud, and ten NA statements e.g. guilty, nervous, hostile. Each statement was measured using a five-point Likert scale ranging from very slightly or not at all to extremely, and then totalled to form overall PA or NA score with higher scores representing higher levels of PA or NA. In the current study, the internal reliability of PANAS was good with a Cronbach's alpha coefficient of .819, and .874 for PA and NA respectively.

## Smartphone Addiction Scale (SAS)

SAS is a 33-item self-report scale used to examine participants' smartphone addiction [33]. SAS contained six sub-factors; daily-life disturbance that measures the extent to which mobile phone use impairs one's activities during everyday tasks (5 statements), positive anticipation to describe the excitement of using phone and de-stressing with the use of mobile phone (8

statements), withdrawal refers to the feeling of anxiety when separated from one's mobile phone (6 statements), cyberspace-oriented relationship refers to one's opinion on online friendship (7 statements), overuse measures the excessive use of mobile phone to the extent that they have become inseparable from their device (4 statements), and tolerance points to the cognitive effort to control the usage of one's smartphone (3 statements). Each statement was measured using a six-point Likert scale from strongly disagree to strongly agree, and total SAS was identified by totalling all 33 statements. Higher SAS scores represented higher degrees of compulsive smartphone use. In the present study, the internal reliability of SAS was identified with Cronbach's alpha correlation coefficient of .918.

## Phone conscious thought and perceived effect on learning

We included a one-item question for phone conscious thought: "During the memory test how often do you think of your smartphone?". The aim of this question was two-fold; first was to capture endogenous interruption experienced by the separation, and second to complement the smartphone addiction to reflect current immediate experience. Participants rated this item on a scale of one (none to hardly) to seven (all the time). We also included a one-item question on how much they perceived their smartphone use has affected their learning and attention: "In general, how much do you think your smartphone affects your learning performance and attention span?". This item was similarly rated on a scale of one (not at all) to seven (very much).

## Procedure

We randomly assigned participants to one of two conditions: low-phone salience (LS) and high-phone salience (HS). Participants were tested in groups of three to six people in a university computer laboratory and seated two seats apart from each other to prevent communication. Each group was assigned to the same experimental condition to ensure similar environmental conditions. Participants in the HS condition were asked to place their smartphone on the left side of the table with the screen facing down. LS participants were asked to hand their smartphone to the researcher at the start of the study and the smartphones were kept on the researcher's table throughout the task at a distance between 50cm to 300cm from the participants depending on their seat location, and located out of sight behind a small panel on the table.

At the start of the experiment, participants were briefed on the rules in the experimental lab, such as no talking and no smartphone use (for HS only). Participants were also instructed to silence their smartphones. They filled in the consent form and demographic form before completing the PANAS questionnaire. They were then directed to CogLab software and began the working memory test. Upon completion, participants were asked to complete the PANAS again followed by the SAS, phone conscious thought, and their perception of their phone use on their learning performance and attention span. The researcher thanked the participants and returned the smartphones (LS condition only) at the end of the task.

## Statistical analysis

We examined for normality in our data using the Shapiro-Wilk results and visual inspection of the histogram. For the normally distributed data, we analysed our data using independent-sample *t*-test for comparison between groups (HS or LS), paired-sample *t* test for within groups (e.g. before and after phone separation), and Pearson *r* for correlation. Non-normally distributed or ranked data were analysed using Spearman rho for correlation.

## Results

### Preliminary analyses

Our female participants reported using their smartphone significantly longer than males, and so we examined the effects of gender on memory recall accuracy. We found no significant difference between males and females on memory recall accuracy, $t$ (117) = .18, $p$ = .86, Cohen's $d$ = .03. Subsequently, data were collapsed, analysed and reported on in the aggregate.

### Smartphone presence and memory recall accuracy

An independent-sample $t$-test was used to examine whether participants' performance on a working memory task was influenced by the presence (HS) or absence (LS) of their smartphone. Results showed that participants in the LS condition had higher accuracy ($M$ = 14.21, $SD$ = 2.61) compared to HS ($M$ = 13.08, $SD$ = 2.53), $t$ (117) = 2.38, $p$ = .02, Cohen's $d$ = .44 (see Fig 1). The effect size $\eta^2$ = .44 indicates that smartphone presence/salience has a moderate effect on participant working memory ability and a sensitivity power of .66.

### Relationship between Smartphone Addiction Score (SAS), higher phone conscious thought and memory recall accuracy

**SAS and memory recall.**   We first examined participants' SAS scores between the two conditions. Results showed no significant difference between the LS (M = 104.64, SD = 24.86) and HS (M = 102.70, SD = 20.45) SAS scores, t (117) = .46, p = .64, Cohen's d = .09. We predicted that those with higher SAS scores will have lower memory accuracy, and thus we examined the relationship between SAS and memory recall accuracy using Pearson correlation coefficient. Results showed that there was no significant relationship between SAS and memory recall accuracy, $r$ = -.03, $n$ = 119, $p$ = .76. We also examined the SAS scores between the LS and HS

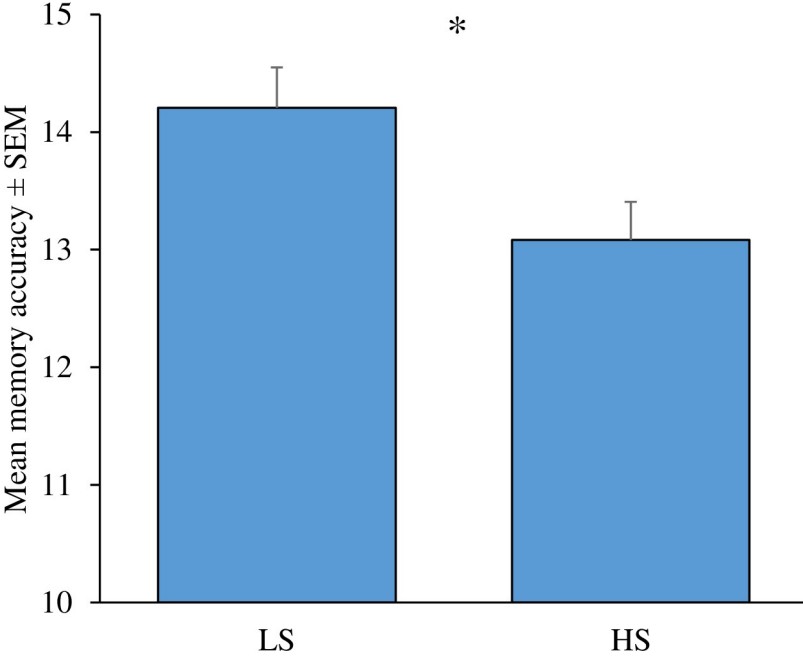

**Fig 1. Mean memory accuracy between low phone salience (LS) and high phone salience (HS) groups (*n* = 119) * *p* < .05.**

groups on memory recall accuracy scores. In the LS group, no significant relationship was established between SAS score and memory accuracy, $r = -.04$, $n = 58$, $p = .74$. Similarly, there was no significant relationship between SAS score and memory accuracy in the HS group, $r = .10$, $n = 61$, $p = .47$. In the event that one SAS subscale may have a larger impact, we examined the relationship between each subscale and memory recall accuracy. Results showed no significant relationship between each sub-factor of SAS scores and memory accuracy, all $ps > .12$ (see Table 2).

**Phone conscious thought and memory accuracy.** We found a significant negative relationship between phone conscious thought and memory recall accuracy, $r_S = -.25$, $n = 119$, $p = .01$. We anticipated a higher phone conscious thought for the LS group since their phone was kept away from them during the task and examined the relationship for each condition. Results showed a significant negative relationship between phone conscious thought and memory accuracy in the HS condition, $r_S = -.49$, $n = 61$, $p = < .001$, as well as the LS condition, $r_S = -.27$, $n = 58$, $p = .04$.

## Affect/mood changes after being separated from their phone

We anticipated that our participants may have experienced either an increase in negative affect (NA) or a decrease in positive affect (PA) after being separated from their phone (LS condition).

We first computed the mean difference (After minus Before) for both positive 'PA difference' and negative affect 'NA difference'. A repeated-measures 2 (Mood change: PA difference, NA difference) x 2 (Conditions: LS, HS) ANOVA was conducted to determine whether there is an interaction between mood change and condition. There was no interaction effect of mood change and condition, $F(1, 117) = .38$, $p = .54$, $n_p^2 = .003$. There was a significant effect of Mood change, $F(1, 117) = 13.01$, $p < .001$, $n_p^2 = .10$ (see Fig 2).

Subsequent post-hoc analyses showed a significant decrease in participants' positive affect before ($M = 31.12$, $SD = 5.79$) and after ($M = 29.36$, $SD = 6.58$) completing the memory task in the LS participants, $t(57) = 2.48$, $p = .02$, Cohen's $d = .28$ but not for the negative affect, Cohen's $d = .07$. A similar outcome was also shown in the HS condition, in which there was a significant decrease in positive affect only, $t(60) = 3.45$, $p = .001$, Cohen's $d = .37$ (see Fig 2).

**PA/NA difference on memory accuracy.** We predicted that LS participants will experience either an increase in NA and/or a decrease in PA since their smartphones were taken away and that this will affect memory recall negatively. Results showed that LS participants who experienced a higher NA difference had poorer memory recall accuracy ($r_s = -.394$, $p = .002$). We found no significant relationship between NA difference and memory recall accuracy for HS participants ($r_s = -.057$, $p = .663$, $n = 61$) and no significant relationship for PA difference in both HS ($r_s = .217$, $p = .093$) and LS conditions ($r_s = .063$, $p = .638$).

**Table 2. Subscales of the Smartphone Addiction Scales (SAS) ($n = 119$).**

| Characteristics | *M* | *SD* | Min | Max | $r_p$ | *p* value |
|---|---|---|---|---|---|---|
| Daily-life disturbance | 16.63 | 5.12 | 5 | 29 | .15 | .11 |
| Positive anticipation | 25.57 | 6.44 | 8 | 42 | -.02 | .80 |
| Withdrawal | 17.63 | 5.52 | 7 | 32 | -.06 | .54 |
| Cyber relation | 18.61 | 5.55 | 7 | 33 | -.04 | .69 |
| Overuse | 15.50 | 4.12 | 4 | 24 | .15 | .12 |
| Tolerance | 9.70 | 3.49 | 3 | 18 | -.01 | .90 |
| Total scores | 103.65 | 22.63 | 45 | 172 | .03 | .76 |

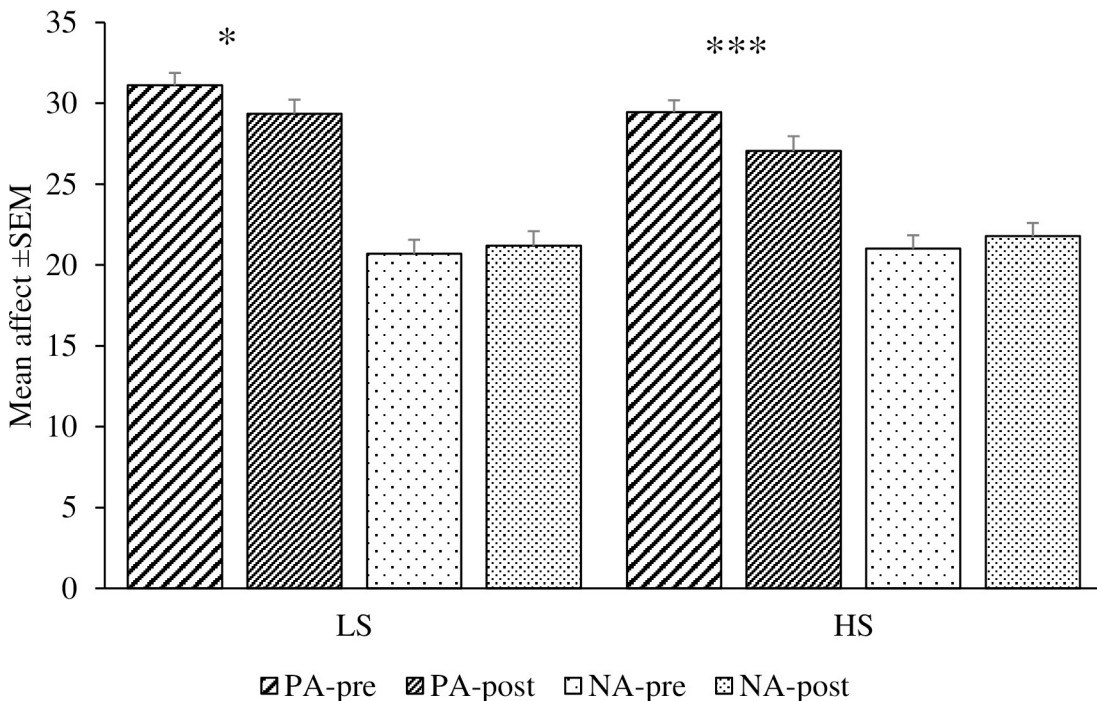

**Fig 2. Mean positive affect (PA) and negative affect (NA) pre- and post-memory task between low phone salience (LS) and high phone salience (HS) groups ($n$ = 119)** *** $p$ < .001, * $p$ < .05.

## Relationship between phone conscious thought, smartphone addiction scale and mood changes to memory recall accuracy

Preliminary analyses were conducted to ensure no violation of the assumptions of normality, linearity, multicollinearity and homoscedasticity. There was a significant positive relationship between SAS scores and phone conscious thought, $r_S$ = .25, $n$ = 119, $p$ = .007. Using the enter method, we found that phone conscious thought explained by the model as a whole was 19.9%, $R^2$ = .20, $R^2_{Adjusted}$ = .17, $F$ (4, 114) = 7.10, $p$ < .001. Phone conscious thought significantly predicted memory recall accuracy, $b$ = -.63, $t$ (114) = 4.76, $p$ < .001, but not for the SAS score, $b$ = .02, $t$ (114) = 1.72, $p$ = .09, PA difference score, $b$ = .05, $t$ (114) = 1.29, $p$ = .20, and NA difference score, $b$ = .06, $t$ (114) = 1.61, $p$ = .11.

## Perception between phone usage and learning

For the participants' perception of their phone usage on their learning and attention span, we found no significant difference between LS ($M$ = 4.22, $SD$ = 1.58) and HS participants ($M$ = 4.07, $SD$ = 1.62), $t$ (117) = .54, $p$ = .59, Cohen's $d$ = .09. There was also no significant correlation between perceived cognitive interference and memory accuracy, $r$ = .07, $p$ = .47.

## Discussion

We aimed [1] to examine the effect of smartphone presence on memory recall accuracy and [2] to investigate the relationship between affective states, phone conscious thought, and smartphone addiction to memory recall accuracy. For the former, our results were consistent with prior studies [11–13] in that participants had lower accuracy when their smartphone was next to them (HS) and higher accuracy when separated from their smartphones (LS). For the

latter, we predicted that the short-term separation from their smartphone would evoke some anxiety, identified by either lower PA or higher NA post-test. Our results showed that both groups had experienced a decrease in PA post-test, suggesting that the reduced PA is likely to have stemmed from the prohibited usage (HS) and/or separation from their phone (LS). Our results also showed lower memory recall in the LS group who experienced higher NA providing some evidence that separation from their smartphone does contribute to feelings of anxiety. This is consistent with past studies in which participants reported increased anxiety over time when separated from their phones [14], or when smartphone usage was prohibited [17].

We also examined another variable–phone conscious thought–described in past studies [11,13], as a measure of smartphone addiction. Our findings showed that phone conscious thought is negatively correlated to memory recall in both HS and LS groups, and uniquely contributed 19.9% in our regression model. We propose that phone conscious thought is more relevant and meaningful compared to SAS as a measure of smartphone addiction [15] because unlike the SAS, this question can capture endogenous interruptions from their smartphone behaviour and participants were to simply report their behaviour within the last hour. The SAS is better suited to describe problematic smartphone use as the statements described behaviours over a longer duration. Further, SAS statements included some judgmental terms such as fretful, irritated, and this might have influenced participants' ability in recalling such behaviour. We did not find any support for high smartphone addiction to low memory recall accuracy. Our participants in both HS and LS groups had similar high SAS scores, and they were similar to Kwon et al. [33] study, providing further evidence that smartphone addiction is relatively high in the student population compared to other categories such as employees, professionals, unemployed. Our participants' high SAS scores and primary use of the smartphone was for social media signals potential problematic users [34]. Students' usage of social networking (SNS) is common and the fear of missing out (FOMO) may fuel the SNS addiction [35]. Frequent checks on social media is an indication of lower levels of self-control and may indicate a need for belonging.

Our results for the presence of a smartphone and frequent phone conscious thought on memory recall is likely due to participants' cognitive load 'bandwidth effect' that contributed to poor memory recall rather than a failure in their memory processes. Past studies have shown that participants with smartphones could generally perform simple cognitive tasks as well as those without, suggesting that memory failure in participants themselves to be an unlikely reason [1,3,5]. Due to our study design, we are unable to tease apart whether the presence of the smartphone had interfered with encoding, consolidation, or recall stage in our participants. This is certainly something of consideration for future studies to determine which aspects of memory processes are more susceptible to smartphone presence.

There are several limitations in our study. First, we did not ask the phone conscious thought at specific time points during the study. Having done so might have determined whether such thoughts impaired encoding, consolidating, or retrieval. Second, we did not include the simple version of this task as a comparison to rule out possible confounds within the sample. We did maintain similar external stimuli in their environment during testing, e.g. all participants were in one specific condition, lab temperature, lab noise, and thereby ruling out possible external factors that may have interfered with their memory processes. Third, the OS task itself. This task is complex and unfamiliar, which may have caused some disadvantages to some participants. However, the advantage of an unfamiliar task requires more cognitive effort to learn and progress and therefore demonstrates the limited cognitive load capacity in our brain, and whether such limitation is easily affected by the presence of a smartphone. Future studies could consider allowing participants to use their smartphone in both conditions and including eye-tracking measures to determine their smartphone attachment behaviour.

### Implications

Future studies should look into the online learning environment. Students are often users of multiple electronic devices and are expected to use their devices frequently to learn various learning materials. Because students frequently use their smartphones for social media and communication during lessons [34,36], the online learning environment becomes far more challenging compared to a face-to-face environment. It is highly unlikely that we can ban smartphones despite evidence showing that students performed poorer academically with their smartphones presented next to them. The challenge is then to engage students to remain focused on their lessons while minimising other content. Some online platforms (e.g. Kahoot and Mentimeter) create a fun interactive experience to which students complete tasks on their smartphones and allow the instructor to monitor their performance from a computer. Another example is to use Twitter as a classroom tool [37].

The ubiquitous nature of the smartphone in our lives also meant that our young graduates are constantly connected to their smartphones and very likely to be on SNS even at work. Our findings showed that the most frequently used feature was the SNS sites e.g. Instagram, Facebook, and Twitter. Being frequently on SNS sites may be a challenge in the workforce because these young adults need to maintain barriers between professional and social lives. Young adults claim that SNS can be productive at work [38], but many advise to avoid crossing boundaries between professional and social lives [39,40]. Perhaps a more useful approach is to recognise a good balance when using SNS to meet both social and professional demands for the young workforce.

## Conclusion

In conclusion, the presence of the smartphone and frequent thoughts of their smartphone significantly affected memory recall accuracy, demonstrating that they contributed to an increase in cognitive load 'bandwidth effect' interrupting participants' memory processes. Our initial hypothesis that experiencing higher NA or lower PA would have reduced their memory recall was not supported, suggesting that other factors not examined in this study may have influenced our participants' affective states. With the rapid rise in the e-learning environment and increasing smartphone ownership, smartphones will continue to be present in the classroom and work environment. It is important that we manage or integrate the smartphones into the classroom but will remain a contentious issue between instructors and students.

## Acknowledgments

We would like to thank our participants for volunteering to participate in this study, and comments on earlier drafts by Louisa Lawrie and Su Woan Wo. We would also like to thank one anonymous reviewer for commenting on the drafts.

## Author Contributions

**Conceptualization:** Clarissa Theodora Tanil.

**Data curation:** Clarissa Theodora Tanil.

**Formal analysis:** Min Hooi Yong.

**Investigation:** Clarissa Theodora Tanil, Min Hooi Yong.

**Methodology:** Min Hooi Yong.

**Resources:** Min Hooi Yong.

**Supervision:** Min Hooi Yong.

**Writing – original draft:** Clarissa Theodora Tanil, Min Hooi Yong.

**Writing – review & editing:** Min Hooi Yong.

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
