## [Decision Letter · Decision Letter 0]

27 Aug 2019

PONE-D-19-17118

Mobile phones: The effect of its presence on learning and memory

PLOS ONE

Dear Dr. Yong, ,

Thank you for submitting your manuscript to PLOS ONE. After careful consideration, we feel that it has merit but does not fully meet PLOS ONE’s publication criteria as it currently stands. Therefore, we invite you to submit a revised version of the manuscript that addresses the points raised during the review process.

Your study addresses an interesting question about the impact of mobile phones on memory.  One area that raised o concerns was your assessment of phone conscious thought.  First you need to provide a clear conceptual  definition of this construct and also your rationale for how to assess it.  In the discussion you seem to imply that phone conscious thought is measuring separation anxiety while there was no assessment of anxiety. Also what is the rationale for measuring affect before and after the memory assessment/?  This point needs to be clarified.    There are also concerns about the analysis of mood changes before and after the memory assessment.  These analyses need to be described more clearly.  Both reviewers raised concerns about your design in terms of your control group.  You need to acknowledge the limitations of your design in the discussion and discuss  how it limits your theoretical interpretation. Overall much more care must be given to the writing of the manuscript.  Reviewer 1 has pointed out numerous examples of how the writing could be improved or clarified.  You must address all points raised by both reviewers in your revised manuscript.  

We would appreciate receiving your revised manuscript by . October 18 2019.  To enhance the reproducibility of your results, we recommend that if applicable you deposit your laboratory protocols in protocols.io, where a protocol can be assigned its own identifier (DOI) such that it can be cited independently in the future. For instructions see: http://journals.plos.org/plosone/s/submission-guidelines#loc-laboratory-protocols

We look forward to receiving your revised manuscript.

Kind regards,

Barbara Dritschel, PhD

Academic Editor

PLOS ONE

Journal Requirements:

2. Please ensure that you have included a section on statistical analysis in your Methods section.

3. Thank you for your ethics statement : "This study was approved by the Department of Psychology Ethics Committee (20171090)."

Reviewers' comments:

Reviewer's Responses to Questions

**Comments to the Author**

1. Is the manuscript technically sound, and do the data support the conclusions?

Reviewer #1: Partly

Reviewer #2: Partly

2. Has the statistical analysis been performed appropriately and rigorously? 

Reviewer #1: No

Reviewer #2: Yes

3. Have the authors made all data underlying the findings in their manuscript fully available?

Reviewer #1: No

Reviewer #2: Yes

4. Is the manuscript presented in an intelligible fashion and written in standard English?

Reviewer #1: Yes

Reviewer #2: No

5. Review Comments to the Author

Reviewer #1: The present study examined the mnemonic consequences associated with the presence of a smartphone. Overall, the authors found that participants without their cellphones had higher accuracy scores than those who had their cell phones present. They also found a negative correlation between accuracy and "phone conscious thought."

Overall, I think this is an interesting area of research. However, the following issues need to be addressed before I can recommend publication. I will start with the larger issues before moving to the smaller issues:

Larger issues

-Probably the biggest issue I found was the interpretation of their results. For example, on pg. 17, the authors state that "Although we did not find a significant relationship between SAS to memory accuracy, our measurements to 'phone conscious thought' is more relevant and meaningful because it measured participants separation anxiety..." This simply cannot be true: First, the question representing" phone conscious thought" asks "During the memory test how often do you think of your smartphone?" What does this even mean, exactly? How did participants interpret this question? Either way, I think it is quite a stretch to consider this anxiety. And, second, the SAS included a "'Withdrawal' sub-factor [that] describ[ed] the feeling of anxiety when separated from one's mobile phone." (pg. 9), but the authors found no significant correlations for any of the subfactors. Thus, not sure how a vague question about thoughts better represents anxiety than the specify subfactor of the SAS.

-Additionally, the authors suggestion that the decrease in positive affect is the result of "prohibited usage/or separation from their phone" (pg. 18). But the authors have no data to support this. For all they know, the participants had a decrease in positive affect simply because they were participating in a study since both groups exhibited this.

-In terms of the procedure, I'm a little concerned that only the "HS group" were told "no phone use." Obviously, I get the logic of this given that the phone was present for them but not for the "LS group." However, this could be a significant confound. Indeed, this could have drawn the participants attention to the fact that they couldn't use it and, in turn, could have distracted them, not simply because it was present but because of the fact that they were told they couldn't use their phone.

-Additionally, did the authors run any preliminary analysed based on how many participants were in each group when they participated? Given the importance just the mere presence of a cell is for the present study, the present of others could have influenced their results as well.

Smaller Issues:

-How is the reader supposed to know what "phone conscious though" means in the abstract?

-Pg. 2, Lines 13-14: A citation is needed to support this.

-Pg. 2 and throughout: "e.g." and "i.e." should only be used in parentheses. Otherwise, it should be "for example" and "that is" respectively and should always have commas around them.

-Pg. 2, Line 19: "Undoubtedly, the constant connectivity is applauded and desired..." This is way too editorial.

-Pg. 3, Line 38: Describe what the "digit cancelation task" is

-Pg. 3, Lines 41-42: "a mobile phone or a phone-sized notebook placed on participant's table before complete the tasks." Is not a complete sentence.

-Pg. 3, Line 42: "...showed no significant on..." Awkward. Reword

-Pg. 3, Line 43: Insert "the" between "during" and "simple"

-Pg. 3, Line 52: "in" should be "on" (there are a lot of typos throughout. I won't highlight them all, but a careful proofreading is necessary

-Pg. 3, Lines 54 & 57: Why do the authors provide the citation number to Ward et al., at the second instance and not the first?

-Pg. 4, Lines 73-78: I think all those sentences could be integrated and stated much more succinctly

-Pg. 5, Line 89 and throughout: The authors use the term "memory" throughout. However, there are many different types of memory. They should specify what they mean exactly by "memory" at each instance.

-Pg. 5, Prior to "present study": I think the authors could do a better job of more explicitly stating what gap in the literature the their study will fill.

-Results: Generally speaking, all t-tests should include cohen's d

-Pg. 7, Line 138: "begun" should be "began."

-Pg. 8, Line 153 & 161: Technically, the 5 should be spelled out. However, at the very least, keep it consistent. That is, the authors us 5 and spell out six.

-Smartphone addiction Scale: Many of the sentences in this section have errors and need to be fixed. Additionally, the authors use "secondly" on line 167, but there's no "first" and there's no "third," etc... Also, examples of each of the sub-factors should be included.

-Pg. 10: Some of this should be in the materials, not the procedure.

-Pg. 11, Gender: Why not include this analysis as a preliminary analysis. If gender, alternatively, is an important issue, then is should be set up as such in the lit review and the authors should examine the interaction with an F-test.

-Pg. 12, Lines 215-220: This should be a preliminary analysis. There's no reason to expect a difference between the groups assuming they were assigned randomly

-Pg. 13 (and elsewhere): The authors sometimes repeat the question in the results. This isn't needed. It's redundant.

-Pg. 14: Why didn't the authors run an ANOVA to examine for an interaction between mood change and condition?

-Pg. 15: More information is needed in terms of the variables included in the model.

-Pg. 18: There are no studies suggested under "Further Studies." The closest is a meaningless sentence: "Future studies should look into the online learning environment."

-Pgs. 18-19: "These behaviors are likely to remain the same when students graduate and move into the workforce." Can the authors provide a citation to back this up or what are the authors basing this on?

-Pg. 19, Lines 327-330: I don't understand this sentence or example...

-Pg. 19, Lines 342-343: "...the extent of the device purpose..." is awkward sounding.

Overall, many typos and awkward phrases. A careful proofreading is necessary.

Reviewer #2: 1. Is the manuscript technically sound, and do the data support the conclusions?

• How was sample size determined? Seems arbitrary, with no power analysis.

• The addition of “phone-conscious thought” is a construct that does not seem to be validated in the peer reviewed literature. It’s ok to include this, but the methods behind the development of these questions should be clearly stated, and the authors must define this construct. There are some problems with how it is defined, because the question used relies specifically on phone-related thoughts during the task, while the phone is either in their presence (HS) or absent (LS). So, this question appears to serve as more of a manipulation check rather than a true measurement of phone-conscious thought. There are many issues with the construct of “phone-conscious thought” in the current manuscript.

• Why is affect measured both before and after the memory test? Explain the rationale. Is the memory test expected to influence mood in any way?

• The inclusion of the phone-conscious thought question in the beginning of the study may have primed participants to think about their phones more overall, and this may have inflated the differences between the LS and HS groups.

• There should have been a 3rd control group where participants were given no instruction about what to do with their phone. This would help assess whether the LS group experienced lower recall or if the HS group experienced higher recall, relative to baseline.

*2. Has the statistical analysis been performed appropriately and rigorously?

• Results for the affect/mood changes are very unclear and should be edited to be more precise. Needs to be much more descriptive.

*3. Have the authors made all data underlying the findings in their manuscript fully available?

Yes

*4. Is the manuscript presented in an intelligible fashion and written in standard English?

The writing is unclear at times with strange vocabulary choices (e.g. “Undoubtedly, the constant connectivity is applauded and desired but this has also spiralled into an obsession with the device for many individuals” lines 19-20). What do the authors mean by “applauded and desired”? Further, writing around the explanations of the relevant literature is imprecise and should be cleaned up so that no previous findings can be mischaracterized. Requires rigorous editing to be publishable, in my opinion.

Lines 57-58: In which direction? And in which tasks? All of them? Needs much greater precision.

6. PLOS authors have the option to publish the peer review history of their article (what does this mean?). If published, this will include your full peer review and any attached files.

Reviewer #1: No

Reviewer #2: No

---

## [Author Response · Author response to Decision Letter 0]

18 Oct 2019

18 October 2019

We would like to express our thanks and gratitude for the helpful comments raised in our paper. Below is a point-by-point response to each comment/question. Please note that the line numbers and pages is taken from the clean version of the revised manuscript. The citations and references are also taken from the clean manuscript, and as such the numbering of the references will be off in this letter. 

Best regards,

C Tanil & MH Yong

---- 

Reviewers' comments:

Reviewer #1: The present study examined the mnemonic consequences associated with the presence of a smartphone. Overall, the authors found that participants without their cellphones had higher accuracy scores than those who had their cell phones present. They also found a negative correlation between accuracy and "phone conscious thought."

Overall, I think this is an interesting area of research. However, the following issues need to be addressed before I can recommend publication. I will start with the larger issues before moving to the smaller issues:

** we thank you for your insightful comments. We have addressed each point in subsequent pages. 

Larger issues

-Probably the biggest issue I found was the interpretation of their results. For example, on pg. 17, the authors state that "Although we did not find a significant relationship between SAS to memory accuracy, our measurements to 'phone conscious thought' is more relevant and meaningful because it measured participants separation anxiety..." This simply cannot be true: First, the question representing" phone conscious thought" asks "During the memory test how often do you think of your smartphone?" What does this even mean, exactly? How did participants interpret this question? Either way, I think it is quite a stretch to consider this anxiety. And, second, the SAS included a "'Withdrawal' sub-factor [that] describ[ed] the feeling of anxiety when separated from one's mobile phone." (pg. 9), but the authors found no significant correlations for any of the subfactors. Thus, not sure how a vague question about thoughts better represents anxiety than the specify subfactor of the SAS.

*** We thank the reviewer for this comment. 

We should explain our reasoning for asking ‘phone conscious thought’ question. In Ward et al.’s study, they included three questions post-task, and we used two out of three questions. The two questions were (1) phone conscious thought “how often were you thinking about your cellphone” and (2) “…to what extent they believed their phones affected their performance and attention spans” (p. 145). The third question was about phone location, and we did not ask this question because we only had two locations and is a pointless question in our study. Ward et al. found that as smartphone salience increases (measured by the 3 questions), available cognitive capacity decreases – which is an indication that this particular question is meaningful to tap endogenous interruptions due to smartphone-related usage throughout the task. Even though the participants in both LS and HS conditions were not allowed to use their phone, their high phone use (average use per day in our sample was 8.16 hours, and 47% participants or 56 out of 119, are considered as addicted when compared to Kwon’s sample) might have evoked such thoughts, as suggested by Wilmer and Chien (2017) in their review. Some participants consider their smartphone as a ‘limb’ and losing this ‘limb’ is more common and has powerful effects than previously thought. 

In Kwon et al.’s paper, the authors described the withdrawal sub-factor as “…involves being impatient, fretful, and intolerable without a smartphone, constantly having one’s smartphone in one’s mind even while not using it, never giving up using one’s smartphone, and becoming irritated when bothered while using one’s smartphone…” (p. 7). The 6 specific questions are as follows: 

1. Won’t be able to stand not having a smartphone 

2. Feeling impatient and fretful when I am not holding my smartphone

3. Having my smartphone in my mind even when I’m not using it

4. I will never give up using my smartphone even when my daily life is already greatly affected by it.

5. Getting irritated when bothered while using my smartphone

6. Bringing my smartphone to the toilet even when I am in a hurry to get there

One of the bigger challenges in using self-reported survey such as SAS is that these questions brings further attention to their behaviour which may then indirectly affects their response behaviour “social desirability” and/or inability to recall the frequency of such behaviour. Having the phone conscious thought is more spot-on and without the risk of both social desirability (negative terms such as impatient, fretful, irritation) and asking individuals to reflect on their past behaviour. 

As to what our participants thought of seeing this question, we think that this is a simple straightforward question. We have since added more information about phone conscious thought in Abstract (page 2, line 6), Introduction (page 6, line 91-97), and Discussion (page 19, line 347-353). 

-Additionally, the authors suggestion that the decrease in positive affect is the result of "prohibited usage/or separation from their phone" (pg. 18). But the authors have no data to support this. For all they know, the participants had a decrease in positive affect simply because they were participating in a study since both groups exhibited this.

*** We thank the reviewer for this comment. Indeed, both LS and HS groups experienced a decrease in PA. We realised that the sentences were misleading, and we apologise for the confusion. We have since reworded the sentences, see below and also on page 20, Line 364-369. 

“While both groups showed a decrease in PA after completing the tasks, it is possible that the reduced PA is likely to have stemmed from the prohibited usage “HS” and/or separation from their phone “LS”. This is consistent with Cheever et al. (15), whose participants reported increased anxiety over time when separated from their phones and with Clayton, Leshner and Almond (18) findings, where participants were unable to use their phone.”

-In terms of the procedure, I'm a little concerned that only the "HS group" were told "no phone use." Obviously, I get the logic of this given that the phone was present for them but not for the "LS group." However, this could be a significant confound. Indeed, this could have drawn the participants attention to the fact that they couldn't use it and, in turn, could have distracted them, not simply because it was present but because of the fact that they were told they couldn't use their phone.

*** Thank you for this comment. The participants were informed to put their phones on silent, and either leave them at their side (HS) or hand them over to the researcher (LS). No phone use instruction was provided to both groups to prevent one group from accessing their phone over another. We have included this confound in the Discussion and suggested improvements. Please see revised section on page 20, line 369-373. 

“Future studies could consider allowing participants to use their phone in both conditions and including eye tracking measures to determine their phone attachment behaviour.”

-Additionally, did the authors run any preliminary analysed based on how many participants were in each group when they participated? Given the importance just the mere presence of a cell is for the present study, the present of others could have influenced their results as well.

*** We thank the reviewer for this comment. We did not analyse for the presence of others as each session was mainly comprised of 3 participants only. We only had 2 sessions of 6 pax per session throughout. 

Smaller Issues:

-How is the reader supposed to know what "phone conscious though" means in the abstract?

*** Thank you for highlighting this. Please see revised section on page 2, line 6. 

-Pg. 2, Lines 13-14: A citation is needed to support this.

*** Thank you. We have added a new citation “GeekWire”. Please see the addition on page 3, line 14. 

-Pg. 2 and throughout: "e.g." and "i.e." should only be used in parentheses. Otherwise, it should be "for example" and "that is" respectively and should always have commas around them.

*** Thank you. We have made the changes throughout. 

-Pg. 2, Line 19: "Undoubtedly, the constant connectivity is applauded and desired..." This is way too editorial.

*** Thank you. We have revised the sentence (see below) and on page 3, line 18-19. 

“Smartphones today have many functions that allows one to be constantly connected to others but this …”

-Pg. 3, Line 38: Describe what the "digit cancelation task" is

*** Thank you for highlighting this omission. We have added the following sentence on page 4, line 40-44. 

“The digit cancellation task involves crossing out one digit from a series of numbers with reference to a target number. Performance is measured by referring to the number of lines completed and a cancellation score based on the total number of targets possible for the lines completed minus the number of errors made (failure to cancel a target or mistakenly cancelled an inappropriate number).” 

-Pg. 3, Lines 41-42: "a mobile phone or a phone-sized notebook placed on participant's table before complete the tasks." Is not a complete sentence.

*** Thank you. We have revised the sentence to “…two groups; a mobile phone or a phone-sized notebook, which were placed on participant’s table before...” Please see page 4, line 46. 

-Pg. 3, Line 42: "...showed no significant on..." Awkward. Reword

*** Thank you. We have revised the sentence to “…significance difference on performance between the phone and notebook condition for the simple digit ….” Please see page 4, line 47. 

-Pg. 3, Line 43: Insert "the" between "during" and "simple"

*** Thank you. Please refer to the above comment as the sentence has been revised. 

-Pg. 3, Line 52: "in" should be "on" (there are a lot of typos throughout. I won't highlight them all, but a careful proofreading is necessary

*** Thank you. We have engaged a native English speaker to proof read our revised manuscript. 

-Pg. 3, Lines 54 & 57: Why do the authors provide the citation number to Ward et al., at the second instance and not the first?

*** Thank you for highlighting this. We have since revised this. 

-Pg. 4, Lines 73-78: I think all those sentences could be integrated and stated much more succinctly

*** Thank you. We have made the changes. Please see page 5, line 77-83. 

-Pg. 5, Line 89 and throughout: The authors use the term "memory" throughout. However, there are many different types of memory. They should specify what they mean exactly by "memory" at each instance.

*** Thank you. We have since included specific types of memory when describing past studies in earlier and subsequent pages. 

-Pg. 5, Prior to "present study": I think the authors could do a better job of more explicitly stating what gap in the literature the their study will fill.

*** Thank you. We have included a research gap in our aim under Present Study. Please see section below, and also found on page 7, line 112-121. 

“Prior studies have demonstrated the detrimental effects of one’s smartphone on cognitive function (e.g. working memory (13), visual spatial search (14), attention (12)), and decreased cognitive ability with increasing attachment to one’s phone (15,17,20). In addition to the presence of a mobile phone , it is also possible that one’s current affective state influences cognitive performance (21–23). But we are uncertain whether one’s current positive or negative affective / mood states plays a bigger role on cognitive function such as memory recall accuracy, suggesting a more complex relationship between current mood states and memory recall accuracy. To our knowledge, no study has examined the relationship between mood states and memory recall accuracy, with smartphone addiction and phone conscious thought as potential mediators. We hypothesised … “

-Results: Generally speaking, all t-tests should include cohen's d

*** Thank you. We have added Cohen’s d for all t-tests. 

-Pg. 7, Line 138: "begun" should be "began."

*** Thank you. We have made the change. 

-Pg. 8, Line 153 & 161: Technically, the 5 should be spelled out. However, at the very least, keep it consistent. That is, the authors us 5 and spell out six.

*** Thank you. We have made the change.

-Smartphone addiction Scale: Many of the sentences in this section have errors and need to be fixed. Additionally, the authors use "secondly" on line 167, but there's no "first" and there's no "third," etc... Also, examples of each of the sub-factors should be included.

 *** Thank you. We have made the changes to include sample questions for each sub-factor. Please see page 10-11, line 190-216. 

-Pg. 10: Some of this should be in the materials, not the procedure.

*** We have since relooked at our procedure and move out some items (e.g. phone conscious thought, and perception on learning) into Materials. 

-Pg. 11, Gender: Why not include this analysis as a preliminary analysis. If gender, alternatively, is an important issue, then is should be set up as such in the lit review and the authors should examine the interaction with an F-test.

*** Thank you for this comment. Gender is not of interest in this study. However, we found that in our sample, females spent more time on their phone compared to males and wanted to determine if there is a gender effect on memory accuracy. We have included a preliminary analysis to include gender analysis under Results section. Please see page 14, line 255-258. 

-Pg. 12, Lines 215-220: This should be a preliminary analysis. There's no reason to expect a difference between the groups assuming they were assigned randomly

*** Indeed, we agree with this comment that there should not be any difference. However, this analysis is more of a precautionary measure. Please see page 14, line 258-261. 

-Pg. 13 (and elsewhere): The authors sometimes repeat the question in the results. This isn't needed. It's redundant.

*** We have now changed our sentences to better reflect our findings. 

-Pg. 14: Why didn't the authors run an ANOVA to examine for an interaction between mood change and condition?

*** Thank you for this suggestion. We have not only added this, but also explained what is PA and NA difference. Please see page 17, line 298-303. 

“We first computed the mean difference (After minus Before) for both positive ‘PA difference’ and negative affect ‘NA difference’. A repeated-measures 2 (Mood change: PA difference, NA difference) x 2 (Conditions: LS, HS) ANOVA was conducted to determine whether there is an interaction between mood change and condition. There was no interaction effect of mood change and condition, F (1, 117) = .38, p = .54, np2 = .003. There was a significant effect of Mood change, F (1, 117) = 13.01, p < .001, np2 = .10 (see Fig 2).”

-Pg. 15: More information is needed in terms of the variables included in the model.

*** Thank you. We have added more information about PA and NA difference score in the earlier results. Please see the above explanation. 

-Pg. 18: There are no studies suggested under "Further Studies." The closest is a meaningless sentence: "Future studies should look into the online learning environment."

*** Thank you. What we meant is actually Implications, rather than Future Studies. We have now reworded the sub-heading. 

-Pgs. 18-19: "These behaviors are likely to remain the same when students graduate and move into the workforce." Can the authors provide a citation to back this up or what are the authors basing this on?

*** We have revised this section. Please refer to page 21, line 387-395.

“The ubiquitous nature of the smartphone in our lives also meant that our young graduates are constantly connected to their phones and very likely to be on SNS even at work. Our findings showed that the most frequently used feature was the SNS sites e.g. Instagram, Facebook, and Twitter. Being frequently on SNS sites may be a challenge in the workforce because these young adults need to maintain barriers between professional and social lives. Young adults claim that SNS can be productive at work (33), but many advise to avoid crossing boundaries between professional and social lives (34,35). Perhaps a more useful approach is to recognise a good balance when using SNS to meet both social and professional demands for the young workforce.”

-Pg. 19, Lines 327-330: I don't understand this sentence or example...

*** We have reworded this section. Please see above. 

-Pg. 19, Lines 342-343: "...the extent of the device purpose..." is awkward sounding.

Overall, many typos and awkward phrases. A careful proofreading is necessary.

*** We have revised it to “… integrate the phones into the classroom but will remain as a contentious issue between… “ See page 21, line 405. 

 

Reviewer #2: 1. Is the manuscript technically sound, and do the data support the conclusions?

• How was sample size determined? Seems arbitrary, with no power analysis.

*** Thank you for this comment. We reported observed power of .66 in our findings, and effect size of ᶇ2 = .44. Please see the added content below and on page 8, line 134-136. 

“Using G*Power v. 3.1.9.2 , we need 76 participants with an effect size of d = .65, α = .05 and power (1-β) = .8 based on Thornton et al.’s study, and 128 participants based on numbers from Ward’s study. “

• The addition of “phone-conscious thought” is a construct that does not seem to be validated in the peer reviewed literature. It’s ok to include this, but the methods behind the development of these questions should be clearly stated, and the authors must define this construct. There are some problems with how it is defined, because the question used relies specifically on phone-related thoughts during the task, while the phone is either in their presence (HS) or absent (LS). So, this question appears to serve as more of a manipulation check rather than a true measurement of phone-conscious thought. There are many issues with the construct of “phone-conscious thought” in the current manuscript.

*** we thank the reviewer for this insightful comment. We certainly did not intend this question to be a manipulation check about their phones. We acknowledged that we omitted a fair amount of phone conscious thought in our earlier submission. We have since added more information about phone conscious thought in Abstract (page 2, line 6), Introduction (page 6, line 91-97), and Discussion (page 19, line 347-353). 

• Why is affect measured both before and after the memory test? Explain the rationale. Is the memory test expected to influence mood in any way?

*** we thank the reviewer for this comment. We realised that this is a major oversight on our part. The main reason for including affect measurement before and after was derived on the possibility that one’s mood may affect your cognitive function, and not simply due to phone presence. We have since made this clearer under Present Study (refer to page 7, line115-121 and in Results (refer to page 16-17, line 295-309). 

• The inclusion of the phone-conscious thought question in the beginning of the study may have primed participants to think about their phones more overall, and this may have inflated the differences between the LS and HS groups.

*** we thank the reviewer for this comment. The phone conscious thought was asked at the end of the memory task. This was included in the Procedure section, page 12, line 243. 

• There should have been a 3rd control group where participants were given no instruction about what to do with their phone. This would help assess whether the LS group experienced lower recall or if the HS group experienced higher recall, relative to baseline.

*** We thank the reviewer for this comment. One of the objectives in this study to examine the effect of a phone presence when participants are completing a simple learning and memory task. For this objective, we had two conditions; phone present (HS) and phone absent (LS). By having a third control with no instructions on what to do with the phone is addressing a different objective and that’s not part of our study objectives. We acknowledged that instruction on phone use may possibly be a confound and as such, we have addressed this limitation in our Discussion (see page 20, line 371-373).

*2. Has the statistical analysis been performed appropriately and rigorously?

• Results for the affect/mood changes are very unclear and should be edited to be more precise. Needs to be much more descriptive.

*** We noted this. We have since revised this section. Please see page 16-17, line 295-309

*3. Have the authors made all data underlying the findings in their manuscript fully available?

Yes

*4. Is the manuscript presented in an intelligible fashion and written in standard English?

The writing is unclear at times with strange vocabulary choices (e.g. “Undoubtedly, the constant connectivity is applauded and desired but this has also spiralled into an obsession with the device for many individuals” lines 19-20). What do the authors mean by “applauded and desired”? Further, writing around the explanations of the relevant literature is imprecise and should be cleaned up so that no previous findings can be mischaracterized. Requires rigorous editing to be publishable, in my opinion.

*** We thank the reviewer for this comment. We have engaged a native English speaker to proofread our manuscript in accordance with academic writing practices. 

Lines 57-58: In which direction? And in which tasks? All of them? Needs much greater precision.

*** we thank the reviewer for this comment. We have revised this sentence to better inform the reader on Ward et al.’s findings.

---

## [Decision Letter · Decision Letter 1]

4 Mar 2020

PONE-D-19-17118R1

Mobile phones: The effect of its presence on learning and memory

PLOS ONE

Dear DR. Yong,

Thank you for submitting your manuscript to PLOS ONE. After careful consideration, we feel that it has merit but does not fully meet PLOS ONE’s publication criteria as it currently stands. Therefore, we invite you to submit a revised version of the manuscript that addresses the points raised during the review process.

I am very sorry for the delay in getting a decision for you.  It was difficult to get reviewers for your paper.  However I now have the response of two reviewers.  The first reviewer thought that your changes made the paper much better.  However there is still a fundamental question about what your paper is addressing.  The second reviewer raised this point as well and made an excellent suggestion that you need to discuss what mechanisms may explain your results and describe how they might be investigated.  You also need to work more on the overall writing style and make sure that the grammar is correct.  If you feel that you can address these issues please submit a revised version of the paper.Please note Reviewer One's points about the role of emotion and discuss how you might investigate its role in a future design as well as why that would be important.  Address all comments raised by the reviewers in your revision or justify why you are not addressing them.

We would appreciate receiving your revised manuscript by April 30 2020. To enhance the reproducibility of your results, we recommend that if applicable you deposit your laboratory protocols in protocols.io, where a protocol can be assigned its own identifier (DOI) such that it can be cited independently in the future. For instructions see: http://journals.plos.org/plosone/s/submission-guidelines#loc-laboratory-protocols

We look forward to receiving your revised manuscript.

Kind regards,

Barbara Dritschel, PhD

Academic Editor

PLOS ONE

Reviewers' comments:

Reviewer's Responses to Questions

**Comments to the Author**

1. If the authors have adequately addressed your comments raised in a previous round of review and you feel that this manuscript is now acceptable for publication, you may indicate that here to bypass the “Comments to the Author” section, enter your conflict of interest statement in the “Confidential to Editor” section, and submit your "Accept" recommendation.

Reviewer #1: (No Response)

Reviewer #3: (No Response)

2. Is the manuscript technically sound, and do the data support the conclusions?

Reviewer #1: Yes

Reviewer #3: Partly

3. Has the statistical analysis been performed appropriately and rigorously? 

Reviewer #1: Yes

Reviewer #3: Yes

4. Have the authors made all data underlying the findings in their manuscript fully available?

Reviewer #1: No

Reviewer #3: Yes

5. Is the manuscript presented in an intelligible fashion and written in standard English?

Reviewer #1: No

Reviewer #3: Yes

6. Review Comments to the Author

Reviewer #1: Overall, I congratulate the authors on the revisions they've made already. The paper is much better for it.

However, I still have some concerns.

Most notably, as far as I can tell, the contribution this manuscript makes to the literature is in the inclusion of emotion. Indeed, the authors make this point quite clear in their "Present Study" section. Indeed, they state "But it is uncertain whether one's current positive or negative affective/mood states plays a bigger role on cognitive function..." but this doesn't seem to cohere very well with the rest of the paper. Barely anything is mentioned about emotion in the intro/lit. review (till the very end). In the analyses, the stats including emotion seem almost like an afterthought. Additionally, emotion is barely mentioned in the discussion. I realize that the authors found no statistical difference across groups, and therefore don't focus on them, but that raises another possible issue: interpreting a null result. If the primary motivation for this study was emotion, it seems to me that one would devise a different design whereby you also manipulate emotion and then examine the different conditions in terms of mobile phone salience.

Thus, at the heart of it, the present paper replicates prior research and then finds a null effect for their primary research question, making interpretations difficult.

For these reasons, I, unfortunately, am recommending rejection.

For the authors reference moving forward, the paper was still a bit hard to parse in places due to language issues throughout. I know the authors state that a native English speaker proofread it, but more diligent proofreading is needed in the future.

Reviewer #3: The experiment presented in this paper is aimed at primarily investigating whether the salience of a phone (high vs. low) impacts memory accuracy. It is a fairly straightforward experimental design and set of results. My main concern is that the paper lacks a clear mechanism to explain the results. Is the main result (i.e., HS leads to lower memory accuracy than LS) due to the fact that high salience participants are distracted during encoding? Is it due to retrieval deficits? Do they not consolidate the information properly? Is it evidence of a bandwidth effect, by which phone-related thought intrusion interferes with memory processes?

My sense is that not only that the experimental design did not attempt to answer the question mechanistically, but there is no attempt to theoretically scaffold the results in a potential mechanism. I would advise the authors to at least speculate as to what could explains the set of results they obtained and to hint at possible investigation of the mechanism involved.

7. PLOS authors have the option to publish the peer review history of their article (what does this mean?). If published, this will include your full peer review and any attached files.

Reviewer #1: No

Reviewer #3: No

---

## [Author Response · Author response to Decision Letter 1]

26 Apr 2020

Reviewer #1: Overall, I congratulate the authors on the revisions they've made already. The paper is much better for it.

However, I still have some concerns.

Most notably, as far as I can tell, the contribution this manuscript makes to the literature is in the inclusion of emotion. Indeed, the authors make this point quite clear in their "Present Study" section. Indeed, they state "But it is uncertain whether one's current positive or negative affective/mood states plays a bigger role on cognitive function..." but this doesn't seem to cohere very well with the rest of the paper. Barely anything is mentioned about emotion in the intro/lit. review (till the very end). In the analyses, the stats including emotion seem almost like an afterthought. Additionally, emotion is barely mentioned in the discussion. I realize that the authors found no statistical difference across groups, and therefore don't focus on them, but that raises another possible issue: interpreting a null result. If the primary motivation for this study was emotion, it seems to me that one would devise a different design whereby you also manipulate emotion and then examine the different conditions in terms of mobile phone salience.

Thus, at the heart of it, the present paper replicates prior research and then finds a null effect for their primary research question, making interpretations difficult.

*** We thank you for this. We do agree that we have omitted a fairly huge amount on the affective state and have since revised the Introduction to explain the interactions between emotion and cognition. Please see Page 4-5, Line 60-73 in Introduction. We have also reorganised the Aim/Hypotheses section – please see Page 5-6, Line 75-92. 

For these reasons, I, unfortunately, am recommending rejection.

For the authors reference moving forward, the paper was still a bit hard to parse in places due to language issues throughout. I know the authors state that a native English speaker proofread it, but more diligent proofreading is needed in the future.

*** We do apologise for this and have since secured a second proof reader. We hope that the manuscript is far more legible now. We have also reorganised parts of the manuscript to make it more coherent. 

----

Reviewer #3: The experiment presented in this paper is aimed at primarily investigating whether the salience of a phone (high vs. low) impacts memory accuracy. It is a fairly straightforward experimental design and set of results. My main concern is that the paper lacks a clear mechanism to explain the results. Is the main result (i.e., HS leads to lower memory accuracy than LS) due to the fact that high salience participants are distracted during encoding? Is it due to retrieval deficits? Do they not consolidate the information properly? Is it evidence of a bandwidth effect, by which phone-related thought intrusion interferes with memory processes?

My sense is that not only that the experimental design did not attempt to answer the question mechanistically, but there is no attempt to theoretically scaffold the results in a potential mechanism. I would advise the authors to at least speculate as to what could explains the set of results they obtained and to hint at possible investigation of the mechanism involved.

*** Thank you for this feedback. We will first address the question on whether the memory recall accuracy was due to encoding, consolidation or retrieval failure. From previous studies, simple versions of a cognitive task was not an issue between low salience (LS) and high salience (HS) participants (1–3) for they had similar performance levels. This suggests it is unlikely that participants had problems at either encoding, consolidation or retrieval for the simple tasks. 

However in our study, we used OS Span task which is considered a complex task compared to simple memory span (4). Although we did not include simple memory span as a contrast to OS Span, previous studies suggest that this is not necessary because of similar performance levels across conditions. One of our aims was to replicate a previous study in investigating whether the presence of a smartphone was sufficient to affect memory recall accuracy (5). We found that our participants had significant difference in memory recall accuracy between HS and LS conditions, p = .02. While our results concurred with previous study findings, we are unable to tease apart whether the presence of the smartphone had interfered with encoding, consolidation, or recall phase in our participants. However there is a possibility that the separation from their smartphones may have caused feelings of anxiety, and anxiety may interrupt memory consolidation as suggested by some (6,7). This is certainly something of consideration for future studies. 

Second, to the bandwidth effect interfering memory processes, we suspect that this might be the case, rather than an issue of failure in a specific memory process. This is because participants with smartphones or texters could generally perform simple cognitive tasks as well as those without, and the presence of the smartphone next to the participant is responsible for the increase in cognitive load (1,3,5). 

Other than the smartphone presence to increase cognitive load, we intended to manipulate participants’ affective state by prohibiting smartphone usage (HS) or taking it away (LS). Previous research has shown that experiencing positive affect (PA) or negative affect (NA) would influence cognitive performance (6–8). We predicted that the short-term separation from smartphone would evoke some anxiety, measured either having lower positive affect (PA) or higher negative affect (NA) post-test. We also predicted that separation from the phone is directly correlated to lower memory recall (LS condition) (part of hypothesis 2). An increase in NA or decrease in PA (as an indicator of separation anxiety to their smartphones) often have a negative effect on cognition (6,7). Further, one study shown an increased level of anxiety even in 10 minutes (9) and OS Span generally takes 20 minutes. Our results supported this hypothesis for LS participants who experienced a stronger negative affect had poorer memory recall accuracy (rs = -.394, p = .002, n = 58). This suggests that phone separation anxiety does increases cognitive load. We did not find any significant relationship between NA and memory recall accuracy for the HS participants and also for the PA difference in both groups (see Results, page 14-15, line 259-265). 

We also examined another variable – phone conscious thought – described in past studies (3,5). Here, we found that phone conscious thought is negatively correlated to memory recall in both HS and LS conditions (see Results page 15, line 273), and uniquely contributed 19.9% in our regression model. 

Taken together, the results showed that phone conscious thought is a significant contributor to the bandwidth effect interrupting their memory processes, and not the change in affective states as we had originally predicted. We do not think that participants’ memory failed at critical points e.g. encoding, retrieval, consolidation. Our participants memory processes are not likely to be impaired as they are neurotypical young adults, unlike well-documented cases in ageing or traumatic brain injury populations. In conclusion, the presence of the smartphone and frequent thoughts of their smartphone were contributors that interrupted their memory processes. 

We do acknowledge several limitations in our study. First, we did not ask the phone conscious thought at specific time points in this study. Having done so might determine whether such thoughts hindered encoding, consolidating, or retrieval. Second, we did not include the simple version of this task as a comparison to rule out possible confounds within the sample. We did maintain similar external stimuli in their environment during testing, e.g. all participants were in one specific condition, lab temperature, lab noise, and thereby ruling out possible external factors that may have interfered with their memory processes. Third, the OS task itself. This task is complex and unfamiliar, thus may have caused some disadvantages to some. However, the advantage of this task being likely to be more unfamiliar – requiring more cognitive effort to learn and progress – demonstrates the limited cognitive capacity in our brain, and whether such limitation is easily affected by a smartphone presence. 

References

1. Ito M, Kawahara J-I. Effect of the presence of a mobile phone during a spatial visual search. Jpn Psychol Res. 2017 Apr 1;59(2):188–98. 

2. Bowman LL, Levine LE, Waite BM, Gendron M. Can students really multitask? An experimental study of instant messaging while reading. Comput Educ. 2010 May 1;54(4):927–31. 

3. Thornton B, Faires A, Robbins M, Rollins E. The mere presence of a cell phone may be distracting: Implications for attention and task performance. Soc Psychol. 2014;45(6):479–88. 

4. Francis G, Neath I, VanHorn D. CogLab On A CD, Version 2.0. Belmont, CA: Wadsworth; 2008. 

5. Ward AF, Duke K, Gneezy A, Bos MW. Brain drain: The mere presence of one’s own smartphone reduces available cognitive capacity. J Assoc Consum Res. 2017 Apr 1;2(2):140–54. 

6. Levine LJ, Lench HC, Karnaze MM, Carlson SJ. Bias in predicted and remembered emotion. Curr Opin Behav Sci. 2018 Feb 1;19:73–7. 

7. Okon-Singer H. The role of attention bias to threat in anxiety: mechanisms, modulators and open questions. Curr Opin Behav Sci. 2018 Feb 1;19:26–30. 

8. Gray JR. Integration of Emotion and Cognitive Control. Curr Dir Psychol Sci. 2004 Apr 1;13(2):46–8. 

9. Cheever NA, Rosen LD, Carrier LM, Chavez A. Out of sight is not out of mind: The impact of restricting wireless mobile device use on anxiety levels among low, moderate and high users. Comput Hum Behav. 2014 Aug 1;37:290–7.

---

## [Decision Letter · Decision Letter 2]

25 Jun 2020

PONE-D-19-17118R2

Mobile phones: The effect of its presence on learning and memory

PLOS ONE

Dear Dr. Yong,

Thank you for submitting your manuscript to PLOS ONE. After careful consideration, we feel that it has merit but does not fully meet PLOS ONE’s publication criteria as it currently stands. Therefore, we invite you to submit a revised version of the manuscript that addresses the points raised during the review process.

I and another reviewer have carefully read your paper and the revisions that you made.  You have addressed many of the points but a few more changes need to be implemented before it can be accepted.  The main issues concern the introduction.  One phrase in Line 62 has no verb and therefore can not be a sentence.  You need to correct the grammatical structure.  Secondly it would be good if the introductory sentence to the paragraph beginning on line 61 indicated the relationship between cognition and affect is important for understanding the impact of mobile phone use on memory.  As it is written the paragraph does link well to the previous paragraphs.  Further line 82 makes reference to smart phone addiction very briefly but the discussion focuses a great deal on smart phone addiction.  You need to define smart phone addiction  and indicate why it is important to examine this construct in your study. Further you need to define the subscales in of the SAS in the methods and also justify under phone consious thought why you are including this question.

We look forward to receiving your revised manuscript.

Kind regards,

Barbara Dritschel, PhD

Academic Editor

PLOS ONE

Reviewers' comments:

Reviewer's Responses to Questions

**Comments to the Author**

1. If the authors have adequately addressed your comments raised in a previous round of review and you feel that this manuscript is now acceptable for publication, you may indicate that here to bypass the “Comments to the Author” section, enter your conflict of interest statement in the “Confidential to Editor” section, and submit your "Accept" recommendation.

Reviewer #3: All comments have been addressed

2. Is the manuscript technically sound, and do the data support the conclusions?

Reviewer #3: Partly

3. Has the statistical analysis been performed appropriately and rigorously? 

Reviewer #3: Yes

4. Have the authors made all data underlying the findings in their manuscript fully available?

Reviewer #3: No

5. Is the manuscript presented in an intelligible fashion and written in standard English?

Reviewer #3: Yes

6. Review Comments to the Author

Reviewer #3: The authors addressed my comments satisfactorily, so I recommend acceptance for the manuscript. I doubt, though, that anxiety plays a huge role in these dynamics, given that it is hard to imagine that one would be able to create the levels of anxiety necessary for the disruption of cognitive function by simply temporarily removing their phones.

7. PLOS authors have the option to publish the peer review history of their article (what does this mean?). If published, this will include your full peer review and any attached files.

Reviewer #3: No

---

## [Author Response · Author response to Decision Letter 2]

14 Jul 2020

1. The main issues concern the introduction. One phrase in Line 62 has no verb and therefore can not be a sentence. You need to correct the grammatical structure. 

*** We thank the editor for pointing this out. We have since removed the sentence. 

2. Secondly it would be good if the introductory sentence to the paragraph beginning on line 61 indicated the relationship between cognition and affect is important for understanding the impact of mobile phone use on memory. As it is written the paragraph does link well to the previous paragraphs. 

*** We thank the editor for this comment. We have revised the paragraph, please see Line 60-61, page 4. 

“Further, we need to consider the relationship between cognition and emotion to understand how frequent mobile phone use affects memory e.g. memory consolidation. Some empirical findings … “ 

3. Further line 82 makes reference to smart phone addiction very briefly but the discussion focuses a great deal on smart phone addiction. You need to define smart phone addiction and indicate why it is important to examine this construct in your study. 

*** We thank the reviewer for omission on our part. Please find the newly added sentences below on Line 83-88, page 5-6.

“One in every four young adults is reported to have problematic smartphone use and this is accompanied by poor mental health e.g. higher anxiety, stress, depression (Sohn et al., 2019). One report showed that young adults reached for their phones 86 times in a day on average compared to 47 times in other age groups (Deloitte Development LLC, 2017). Young adults also reported that they “definitely” or “probably” used their phone too much, suggesting that they recognised their problematic smartphone use. “

4. Further you need to define the subscales in of the SAS in the methods and also justify under phone consious thought why you are including this question.

*** We thank the reviewer for this comment. Please see the inclusion for SAS subscales on Line 159-166, page 9. 

“SAS contained six sub-factors; daily-life disturbance that measures the extent to which mobile phone use impairs one’s activities during everyday tasks (5 statements), positive anticipation to describe the excitement of using phone and de-stressing with the use of mobile phone (8 statements), withdrawal refers to the feeling of anxiety when separated from one’s mobile phone (6 statements), cyberspace-oriented relationship refers to one’s opinion on online friendship (7 statements), overuse measures the excessive use of mobile phone to the extent that they have become inseparable from their device (4 statements), and tolerance points to the cognitive effort to control the usage of one’s smartphone (3 statements).”

We have also added the justification to the phone conscious thought. Please see this inclusion on Line 173-175, page 9. 

“The aim of this question was two-fold; first was to capture endogenous interruption experienced by the separation, and second to complement the smartphone addiction to reflect current immediate experience.” 

Reviewer #3: The authors addressed my comments satisfactorily, so I recommend acceptance for the manuscript. I doubt, though, that anxiety plays a huge role in these dynamics, given that it is hard to imagine that one would be able to create the levels of anxiety necessary for the disruption of cognitive function by simply temporarily removing their phones.

*** 

5. How frequent do we reach for our phones? 

 *** Surprisingly, high, but unsurprisingly higher for young adults. Deloitte 2017 survey reported that the average American reaches for their phones 47 times while young adults (aged between 18 to 24) reach for 86 times. The same survey also reported that 89% looked at their phone within an hour of waking up and that 81% also looked at their phone within an hour before going to bed. Further, 90% of young adults reported using their phones in their daily activities ranging from shopping, leisure time, talking to friends, crossing the road, and this trend has been consistent for the past three years. The young adults also reported that they “definitely” or “probably” use their phone too much, suggesting some form of recognising their addiction (Deloitte Development LLC, 2017). Another poll reported that one in every 10 Americans check their phones every four minutes, and that most people struggle to go beyond 10 minutes without checking their phone (SWNS, 2017). 

We have added couple of sentences to further highlight on mobile phone addiction under Research Aim (see Line 85-88, page 5-6).

---

## [Editor Report · Decision Letter 3]

3 Aug 2020

Mobile phones: The effect of its presence on learning and memory

PONE-D-19-17118R3

Dear Dr. Yong,

We’re pleased to inform you that your manuscript has been judged scientifically suitable for publication and will be formally accepted for publication once it meets all outstanding technical requirements.

Kind regards,

Barbara Dritschel, PhD

Academic Editor

PLOS ONE
---

## [Editor Report · Acceptance letter]

6 Aug 2020

PONE-D-19-17118R3 

Mobile phones: The effect of its presence on learning and memory 

Dear Dr. Yong:

I'm pleased to inform you that your manuscript has been deemed suitable for publication in PLOS ONE. Congratulations! Your manuscript is now with our production department. 

Kind regards, 

on behalf of

Dr. Barbara Dritschel 

Academic Editor

PLOS ONE